# SMOFlipid Impact on Growth and Neonatal Morbidities in Very Preterm Infants

**DOI:** 10.3390/nu14193952

**Published:** 2022-09-23

**Authors:** Suzan S. Asfour, Belal Alshaikh, Latifah AlMahmoud, Haider H. Sumaily, Nabeel A. Alodhaidan, Mousa Alkhourmi, Hissah A. Abahussain, Thanaa M. Khalil, Bushra A. Albeshri, Aroub A. Alhamidi, Maha R. Al-Anazi, Raneem S. Asfour, Mountasser M. Al-Mouqdad

**Affiliations:** 1Clinical Pharmacy Department, Pharmaceutical Care Services, King Saud Medical City, Riyadh 12746, Saudi Arabia; 2Department of Pediatrics, Cumming School of Medicine, University of Calgary, Calgary, AB T2N 1N4, Canada; 3Neonatal Intensive Care, Hospital of Pediatrics, King Saud Medical City, Riyadh 12746, Saudi Arabia; 4Pediatric Gastroenterology Department, Hospital of Pediatrics, King Saud Medical City, Riyadh 12746, Saudi Arabia; 5General Pediatrics Department, Hospital of Pediatrics, King Saud Medical City, Riyadh 12746, Saudi Arabia; 6Obstetric and Gynecology Department, Maternity Hospital, King Saud Medical City, Riyadh 12746, Saudi Arabia; 7Pharmacy Department, Pharmaceutical Care Services, King Saud Medical City, Riyadh 12746, Saudi Arabia; 8Pharmacy Department, Pharmaceutical Care Services, Ministry of Health, Riyadh 12613, Saudi Arabia; 9Pharmacy College, Jordan University of Science and Technology, Irbid P.O. Box 3030, Jordan

**Keywords:** SMOFlipid, intralipid, growth, morbidities, mortality, preterm infants

## Abstract

The soybean oil, medium-chain triglycerides, olive oil, and fish oil lipid (SMOFlipid) is increasingly being used worldwide without definite evidence of its benefits. We examined the effect of SMOFlipid on growth velocity and neonatal morbidities in very preterm infants. Very preterm infants who received soybean-based lipid emulsion between January 2015 and 2018 were compared with those who received SMOFlipids between 2019 and January 2022 in our neonatal tertiary center. Linear regression analysis was conducted to analyze the association between type of lipid emulsion and growth velocity. Modified log-Poisson regression with generalized linear models and a robust variance estimator (Huber–White) were applied to adjust for potential confounding factors. A total of 858 infants met our inclusion criteria. Of them, 238 (27.7%) received SMOFlipid. SMOFlipid was associated with lower growth velocity between birth and 36-week corrected gestational age compared with intralipid Δ weight z-score (adjusted mean difference (aMD) −0.67; 95% CI −0.69, −0.39). Subgroup analysis indicated that mainly male infants in the SMOFlipid–LE group had a lower Δ weight z-score compared to those in the intralipid group (*p* < 0.001), with no difference observed in females (*p* = 0.82). SMOFlipid was associated with a lower rate of bronchopulmonary dysplasia (BPD) (aRR 0.61; 95% CI 0.46, 0.8) and higher rate of late-onset sepsis compared with intralipid (aRR 1.44; 95% CI 1.22–1.69). SMOFlipid was associated with lower growth velocity and BPD but higher rate of late-onset sepsis—it is a double-edged sword.

## 1. Introduction

Very preterm infants are born with insufficient energy stores and underdeveloped gastrointestinal tract and immune systems [1]. They remain at high risk of postnatal growth failure [2]. Parenteral nutrition (PN) is the main early source of energy and is the standard of care in preterm infants in the first days of life [3]. Early PN contains amino acid, dextrose, and intravenous lipid. Intralipid provides up to 50% nonprotein energy, essential fatty acids, and very-long-chain polyunsaturated fatty acids [4,5]. Globally, different lipid emulsions are used, and most of them contain soybean oil, which is a rich source of the essential omega-6 polyunsaturated fatty acid (ω-6PUFA) that in turn plays a critical role in developing PN-associated liver disease and promoting inflammation and immune suppression [6,7,8,9].

A new generation of lipid emulsions has been widely introduced. It comprises a mixture of 30% soybean oil, 30% medium-chain triglycerides (MCTs), 25% olive oil, and 15% fish oil: the so-called SMOFlipid (Fresenius Kabi). The major target of SMOFlipid is to reduce the soybean oil’s detrimental effects. A recent meta-analysis showed that SMOFlipid reduces the liver injury and hyperlipidemia and shortens the period of hospital stay; however, it could not reduce the important inflammatory markers in adults who are admitted for surgical reasons [10]. Another study found that SMOFlipid decreases the proportion of several essential fatty acids, including linoleic acid (LA) and α-linolenic acid (ALA) [10,11].

In the pediatric population, the Congress of Intestinal Rehabilitation and Transplantation showed that the trend of SMOFlipid use is moving up as the first-line choice for pediatric patients [7]. A recent study found that SMOFlipid improves the lipid panel in patients with intestinal failure. It increases ALA, LA, and docosahexaenoic acid (DHA) [12]. Another study on pediatric patients receiving home PN showed that SMOFlipid does not change the lipid profile but decreases the serum bilirubin [12,13].

Despite limited evidence, use of SMOFlipid is increasing globally. A recent randomized controlled trial found that the antioxidant capacity was lower in preterm infants who received SMOFlipid than in other infants who received another intralipid form (ClinOleic) [14]. It also reduced the severe bronchopulmonary dysplasia (BPD) rate but did not change the BPD, intraventricular hemorrhage (IVH), and neonatal sepsis incidence rates [14]. Other recent studies showed that SMOFlipid might reduce the severity of retinopathy of prematurity (ROP) and cholestasis, but the severe BPD and mortality rates do not alter [15]. Regarding essential fatty acid, a recent randomized controlled trial showed that SMOFlipid decreases the arachidonic acid to DHA ratio in preterm infants [16,17]. With regard to the influence of parenteral lipid on anthropometric measurements, there is overall consensus that the delay in introducing lipid in the first days of life in preterm infants is associated with slow growth velocity, although the results are conflicting whether SMOFlipid has beneficial or harmful effects on growth [18,19,20].

The aim of our study is to investigate the SMOFlipid impact on growth velocity, major neonatal outcomes, and preterm infants’ mortality.

## 2. Methods

### 2.1. Study Design

This retrospective chart review included a cohort of preterm infants who were admitted to the neonatal intensive care unit (NICU) of the King Saud Medical City (KSMC) tertiary referral center between January 2015 and January 2022.

Including level 3, the NICU at KSMC has an average annual admission of 1100 patients. This study was conducted in accordance with the Declaration of Helsinki and Good Pharmacoepidemiology Practice Guidelines and was approved by the medical ethical review committee of KSMC. The requirement for consent was waived.

### 2.2. Inclusion and Exclusion Criteria

We included very preterm infants who were born at KSMC at ≤32 weeks of gestation, had a birth weight of <1500 g, and were admitted to the NICU. All infants who received total PN plus lipid emulsion within the first 24 h of birth were included in our study. We excluded infants with major congenital anomalies or congenital infection, inborn errors of metabolism, those who did not receive PN, those who were not born at KSMC or were transferred to another hospital or died within the first 7 days, and those whose data could not be retrieved.

### 2.3. Data Collection and Follow-Up

The infants’ charts from NICU admission until discharge or death were reviewed. Demographic, clinical, and outcome data, including the major morbidities associated with prematurity, were obtained. Maternal data, including gestational diabetes mellitus, maternal hypertension, antenatal steroid treatment, and mode of delivery, were also retrieved. The details about the measurement of body weight, head circumference (HC), and lipid profile are provided in Section 2.5.1, Section 2.5.2, Section 2.5.3 and Section 2.6.

### 2.4. Study Outcome

The primary outcome of this study was the change in weight z-score (weight Δ z-score) from birth to 36 weeks of post-menstrual age (PMA) or at discharge, whichever one came first during two different period changes when SMOFlipid was routinely included in PN formulation.

Secondary outcomes were the change in HC z-score (HC Δ z-score) from birth to 36 weeks of PMA or at discharge, discharge weight, z-score of discharge weight, days of PN and lipid emulsion (LE), and length of hospital stay, as well as the incidence of mortality and preterm neonatal morbidities including BPD, late onset of sepsis (LOS), ROP, any or severe intraventricular hemorrhage (IVH), and necrotizing enterocolitis (NEC).

### 2.5. Definitions

#### 2.5.1. Nutrition Protocol

Parenteral nutrition (PN): The PN was started early after birth using starter PN. Individualized PN was prescribed daily. Starter PN contains dextrose 10%, amino acids 4%, and calcium gluconate 0.01 mmol/mL. Individualized PN solution containing amino acids, glucose, minerals, trace elements, water-soluble vitamins, and fat-soluble vitamins was started within the first 24 h of life and infused continuously for 24 h.

The subjects were categorized into two groups of LE—(1) SMOFlipid (Fresenius Kabi, Melrose Park, IL, USA): infants were given parenteral multi-oil emulsions, containing soybean oil, MCT, olive oil, and fish oil; and (2) intralipid 20% medium-chain and long-chain fat injection: containing soybean oil and MCT.

The initial LE dose was 0.5 to 1.0 g per kg per day early after birth and was increased by 0.5 to 1.0 g per kg per day every 24 h with a maximum of 3.0 g to 3.5 g lipids per kg per day. Triglyceride level was not measured in the routine investigation. Minimal enteral nutrition was started as soon as possible after birth if possible. A preterm formula or expressed breast milk was administrated through an orogastric tube intermittently according to the feeding protocol, which depends on birth weight.

Preterm formula was administered when human milk was not available or sufficient.

#### 2.5.2. Growth Anthropometry

The body weight and HC were recorded when entering the neonatal unit. Body weight and HC was measured by NICU nurses. The body weight was measured with an electronic scale calibrated to 0.05 kg. The HC measurement was obtained weekly with a measuring tape, which is precise to the nearest millimeter. The change in body weight was recorded daily, with HC weekly records. For the z-score calculation, the 2013 Fenton growth chart was used [21]. Postnatal growth velocities were defined as changes in z-scores (Δ weight z, Δ HCZ) to 36 weeks of PMA or at discharge, whichever came first.

#### 2.5.3. Preterm Neonatal Morbidities

Diagnoses of IVH, BPD, NEC, as well as sepsis were retrieved from clinical records. Severe IVH was defined as grade III–IV IVH on Papile classification [22]. BPD was defined as the need for supplemental oxygen at 36-week PMA [23]. The NEC was diagnosed by clinical signs and radiologic findings (Bell’s stages 2–3) [24]. Sepsis was diagnosed by clinical symptoms accompanied by a positive blood culture [25]. ROP was classified according to the International Classification of Retinopathy of Prematurity [26]. Severe ROP was defined as stage 3 or more. Growth velocity was calculated using the following equation = [1000 × ln (discharge Weight/birth Weight)]/length of hospital stay, where ln is the natural logarithm, weights are expressed in grams, and length of hospital stay in days.

### 2.6. Statistical Analysis

Before performing the analysis, we checked the dataset for missing data. Data were analyzed using a statistical software package (Statistical Package for the Social Sciences, version 25.0, SPSS Inc., Chicago, IL, USA).

Data regarding maternal and infant variables were presented using descriptive statistics, including median, interquartile range (IQR), frequency, and percentage. Fisher’s exact test was used to determine the association between categorical variables. The Mann–Whitney U test was used for between-group comparisons of ordinal qualitative variables. For between-group comparisons of continuous variables, the unpaired Student’s *t*-test was used for normally distributed data, and the Mann–Whitney U test was used for non-normally distributed data. The Kolmogorov–Smirnov test and a visual inspection of histograms were performed to evaluate the distribution of quantitative variables.

To analyze the association between type of LE and outcomes, we first conducted a univariate relative risk analysis on the recorded variables (gestational age; small for gestational age; gender; 5 min Apgar score; necrotizing enterocolitis; surfactant use; maternal hypertension; antenatal and postnatal steroid treatment; premature rupture of membrane, late onset of sepsis; delivery mode; dextrose intake; amino acids; and LE) because we considered them to be potential confounders. All factors with a *p* value of <0.05 in the univariate analysis were included in the final multivariable regression model. Linear regression analysis was conducted to analyze the association between type of LE and growth anthropometrics after checking for collinearity with a correlation matrix. Modified log-Poisson regression with generalized linear models and a robust variance estimator (Huber–White) were applied for univariate relative risk analysis and to the models to adjust the relative risk for neonatal morbidities and mortality. The BPD mediation effect and grades 3 or 4 IVH on weight velocity and weight, length, and HC at 36-week PMA were evaluated using a causal mediation analysis. All statistical tests were two-tailed, and *p* values of <0.05 were considered statistically significant.

## 3. Results

During the study period, 1976 preterm infants with ≤32 gestational weeks and birth weight <1500 g were admitted to the NICU (level 3). Of them, 858 met the inclusion criteria and were eligible for inclusion in the final analysis (Figure 1).

Among 858 infants, 238 received SMOFlipid and 620 were in the intralipid group. Maternal and neonatal characteristics are summarized in Table 1. There were higher percentage of cesarean deliveries in the SMOFlipid–LE group compared with the intralipid group. The Apgar scores at 1 and 5 min were higher in infants who received SMOFlipid compared with the intralipid group. In addition, infants who received SMOFlipid received more surfactant and required mechanical ventilators more often compared with the intralipid group (Table 1).

Univariate analysis showed that median intake of macronutrients including parenteral lipid (g/kg/day) and protein (g/kg/day) either the total intake or in the first postnatal week was significantly higher in the SMOFlipid group compared with the intralipid group (*p* < 0.001). In contrast, the median of parenteral carbohydrate intake, either the total intake or the first postnatal week (mg/kg/min), was lower in the SMOFlipid compared with intralipid group (*p* < 0.001). We observed no association between energy intakes and type of LE. Infants who received expressed breast milk were higher in SMOFlipid compared with the intralipid group (*p* < 0.001) (Table 2).

Infants who received SMOFlipid independently and negatively affected the Δ weight z-score compared with the intralipid group (*p* < 0.001). In addition, the growth rate and z-score at discharge weight were significantly higher in infants who received intralipid compared with infants who received SMOFlipid (*p* = 0.02) (Table 3). The percentile weight changes in infants stratified by types of LE are demonstrated in Figure 2. Additionally, in the univariate analysis, we found that infants who received SMOFlipid were significantly associated with lower BPD and higher LOS compared with infants who received intralipid (*p* < 0.001) (Table 3).

The multivariable regression analysis performed after adjusting the variables that were significant in the univariate analysis revealed that the SMOFlipid administration had a negative impact on Δ weight z-score compared with intralipid (aRR −0.67; 95% CI −0.69 to −0.39) (Table 4). The SMOFlipid effect on Δ weight z-score was also different according to sex (interaction *p* = 0.001). Males in the SMOFlipid–LE group had a lower Δ weight z-score compared to those in the intralipid group (*p* < 0.001), with no difference observed in females (*p* = 0.82) (Table 5). Causal mediation analyses showed a negative BPD effect on Δ weight z-score for males and females (indirect effect for males, −0.87, 95% CI: −1.23 to −0.50, *p* < 0.001; and indirect effect for females, −0.1.23, 95% CI: −1.63 to −0.83, <0.001). However, grade 3 or 4 IVH did not have a mediating effect for either male and female infants (indirect effect for males, −0.03, 95% CI: −0.46 to 0.41, *p* = 0.91; and indirect effect for females, −0.07, 95% CI: −0.44–0.57, *p* = 0.8).

Furthermore, the SMOFlipid administration compared to the intralipid group significantly reduced the BPD (aRR 0.61; 95% CI 0.46–0.8). However, infants who received SMOFlipid had significantly higher LOS compared with those who received intralipid (aRR 1.44; 95% CI 1.22–1.69).

However, aRR revealed that there was no association between SMOFlipid and neonatal morbidity (severe IVH, PDA, NEC, LOHS, PN duration, and lipid duration) as well as neonatal mortality.

## 4. Discussion

Our study indicates that SMOFlipid is associated with low growth velocity in very preterm male infants. In addition, it is associated with increased proportion of late onset of sepsis and decreased rate of BPD.

In terms of growth velocity, the main target of PN is to meet the nutritional requirements in preterm infants and achieve weight gain like a normal fetus [27]. Lipid is composed of the majority source of nonprotein energy and essential fatty acids. Vlaardingerbroek et al. found on their randomized controlled trial that SMOFlipid has a positive impact on the anthropometry of preterm infants [19]. However, other studies including clinical trials demonstrated that SMOFlipid emulsion does not improve the weight z-score and HC [20,28]. Other trials showed SMOFlipid’s negative impact on growth parameters [29,30]. Our study manifests that SMOFlipid is associated with delay in Δ weight z-score. The delayed growth attributes to the low levels of certain essential fatty acids such ARA and LA. All studies, including our study, present nonhomogeneous findings. This discrepancy represents that optimal infant growth needs the presence of all essential fatty acids, including ARA, DHA, and EPA. More than that, any change in the percentage of their presence and ratios will affect the growth rates [31,32]. Furthermore, SMOFlipid administration is associated with delayed growth velocity, mainly in males. This effect may have been mediated by BPD. In its secondary analysis, the MOBYDIck trial found that giving DHA supplementation alone to mothers who breastfed their preterm infants is associated with improving the weight velocity in females, only with an opposite impact on males, and these influences are not mediated by BPD [33].

Our resulted showed increased risk of late-onset sepsis. A decade ago, meta-analysis showed that SMOFlipid emulsions are weakly associated with fewer episodes of late onset of sepsis compared with soybean oil emulsions (RR: 0.75; 95% CI: 0.56, 1.00) [34]. They explained their findings that excess intake of soybean-based LE increases the pro-inflammatory eicosanoids and oxidative stress [35,36]. A trial which was conducted later by the same team of the meta-analysis found no association between the neonatal sepsis rate and the kind of LE [19]. A more recent trial found that SMOFlipid reduces the ratio of ARA to DHA [16]. These findings were supported by earlier animal studies that found that increasing the DHA level will disturb the ARA level [37]. The ARA plays a fundamental role in regulating the immune and inflammatory responses; therefore, any disturbance in its level may predispose the immature infant to develop late onset of sepsis [38].

Our study showed that preterm infants who received SMOFlipid as part of their PN in their early life have a lower BPD rate. The DHA, as part of SMOFlipid, stimulates surfactant production and minimizes the inflammatory process in animals’ lungs [39,40,41,42]. Not all studies on preterm infants displayed the same results. The DINO trial showed that DHA supplements reduce the BPD incidence in preterm infants [43]. In contrast, other trials showed different outcomes [44,45]. Both the N3RO trial and MOBYDIck trial demonstrated no change in the BPD rate after applying DHA supplements either directly through enteral feeding to preterm infants or through providing them orally to pregnant women. Two recent meta-analyses indicate omega-3 long-chain polyunsaturated fatty acids do not reduce the BPD proportion in preterm infants [46,47]. There is a trial now underway investigating whether parenteral omega-3-enriched LE reduces the BPD incidence in preterm infants who are less than 30 weeks gestational age [48].

With regard to other neonatal morbidities, such as ROP, IVH, NEC, and mortality, our study did not find any potential benefit of SMOFlipid in reducing their incidence rates. Previous studies demonstrated conflicting results. The recent meta-analysis found that DHA supplements do not make any significant changes in NEC, IVH, and severe ROP [46]. However, another meta-analysis found fish oil lipid emulsion is effective in reducing the ROP incidence and severity [49].

Current studies present conflicting data regarding the short-term and long-term impacts of SMOFlipid in preterm infants. These differences might be attributed to the population enrolled in each study, the quantity of the supplements, and the method of preparation and giving they received, such as whether they received DHA only or lipid that is enriched with DHA, in addition to other components.

Several limitations are present in the current study. Our study is retrospective in nature and not all confounding factors may have been adjusted for. We did not measure the essential fatty acid levels, as it is not a routine test, and we perform it for preterm infants unless there are indications of metabolism inborn error. Furthermore, long-term follow-up would have strengthened the study. We have good sample size, and we were able to investigate the important neonatal outcomes and controversies at the same time. We were able to highlight the present conflict in the current practice of SMOFlipid in preterm infants.

## 5. Conclusions

In conclusion, our study revealed that none of the studied lipid emulsions were proven to be superior to the others for very preterm infants. SMOFlipid may reduce the risk of BPD but increases the risk of late onset of sepsis and postnatal growth failure. A large randomized controlled trial is required to identify effects of SMOFlipid on neonatal morbidities.

## Figures and Tables

**Figure 1 nutrients-14-03952-f001:**
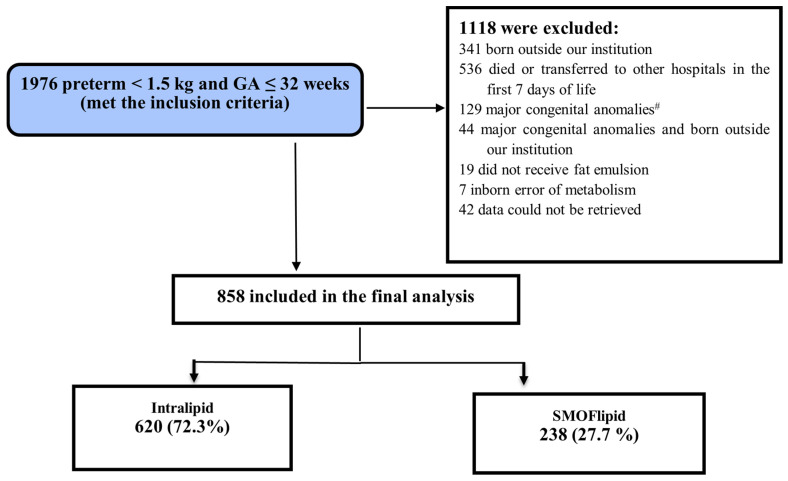
Flow chart of study participants.

**Figure 2 nutrients-14-03952-f002:**
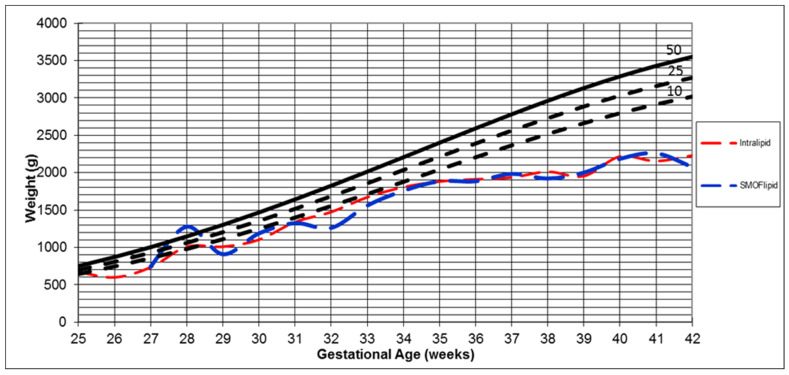
Growth of preterm infants who received SMOFlipid vs. intralipid. The red dash indicates the relationship between the average weight and gestational age among infants who received intralipid. The blue dash indicates the relationship between the average weight and gestational age among infants who received SMOFlipid.

**Table 1 nutrients-14-03952-t001:** Maternal and neonatal characteristics of study participants with SMOFlipid and intralipid.

Variable	N	Intralipid (n = 620)	SMOFlipid (n = 238)	*p* Value
Gestational age (weeks), median (IQR)	858	29 (27.0–30.8)	29 (27.0–31.0)	0.12
Birth weight (grams), (IQR)	858	1100 (875–1344)	1100 (870–1310)	0.40
Length (cm), median (IQR)	858	37 (34–39)	37 (33–39)	0.4
Head circumference (cm), (IQR)	858	26 (24–28)	27 (24–28)	0.25
Small for gestational age, n (%)	858	38 (6.1)	22 (9.2)	0.13
1 min Apgar score, median (IQR)	858	5 (3–6)	6 (4–7)	0.006 *
5 min Apgar score, median (IQR)	858	7 (6–8)	7 (7–8)	<0.001 *
Male, n (%)	858	320 (51.6)	126 (52.7)	0.82
Antenatal steroid treatment, n (%)	858	329 (53.1)	130 (54.4)	0.76
Gestational diabetes mellitus, n (%)	858	27 (4.4)	14 (5.9)	0.37
Maternal hypertension, n (%)	858	146 (23.5)	62 (25.9)	0.48
Preterm rupture of membrane, n (%)	858	77 (12.4)	18 (8)	0.08
Cesarean section, n (%)	858	251 (40.5)	169 (70.7)	<0.001 *
Inotropes (dopamine, dobutamine, epinephrine, milrinone), n (%)	858	237 (38.2)	104 (43.5)	0.16
Hydrocortisone, n (%)	858	124 (20)	53 (22.2)	0.51
Omeprazole, n (%)	858	66 (10.6)	19 (7.9)	0.25
Noninvasive respiratory support, n (%)	858	519 (83.7)	207 (86.6)	0.34
Respiratory distress syndrome required surfactant, n (%)	858	380 (61.3)	174 (72.8)	0.001 *
Mechanical ventilation, n (%)	858	426 (68.7)	187 (78.2)	0.005 *
Patent ductus arteriosus required treatment, n (%)	858	49 (7.9)	25 (10.5)	0.23
Peripherally inserted central catheter (PICC), n (%)	858	196 (31.6)	100 (41.8)	0.005 *
Corrected gestational age at discharge	661	36 (34-38)	36 (33-38)	0.55

* Statistically significant at 5% level.

**Table 2 nutrients-14-03952-t002:** Nutritional and growth characteristics of study participants with SMOFlipid and intralipid.

Variable	N	Intralipid (n = 620)	SMOFlipid (n = 238)	*p* Value
Parenteral nutrition				
Duration of parenteral nutrition (days), median (IQR)	858	16 (8–32)	13 (7–26)	0.14
Duration of parenteral lipid (days), median (IQR)	858	7 (5–7)	7 (5–7)	0.07
Average parenteral lipid intake (g/kg/day), median (IQR)	858	2 (1.6–2.4)	2.2 (1.8–2.6)	<0.001 *
Average parenteral protein intake (g/kg/day), median (IQR)	858	3.8 (3.6–4)	4 (3.9–4)	<0.001 *
Average parenteral carbohydrate intake (mg/kg/min), median (IQR)	858	8.5 (7.9–9.2)	8.2 (7.2–8.7)	<0.001 *
Average parenteral calorie intake (kcal/kg/day), median (IQR)	858	76.42 (68.74–82.06)	75.48 (68.11–80.85)	0.15
Average parenteral lipid intake in the 1st 7 days (g/kg/day), median (IQR)	858	1.8 (1.4–2.3)	2.2 (1.8–2.6)	<0.001 *
Average parenteral protein intake in the 1st 7 days (g/kg/day), median (IQR)	858	4 (3.6–4)	4 (4–4)	<0.001 *
Average parenteral carbohydrate intake in the 1st 7 days (mg/kg/min), median (IQR)	858	8.2 (7.5–8.8)	7.6 (6.8–8.2)	<0.001 *
Average parenteral calorie intake in the 1st 7 days (kcal/kg/day), median (IQR)	858	73.03 (61.57–79.93)	73.85 (65.12–79.71)	0.56
Enteral nutrition				
Age of starting feeding (days), median (IQR)	858	3 (1–5)	3 (1–5.25)	0.15
Received expressed breastmilk, n (%)	858	304 (49)	155 (64.9)	<0.001 *
High calorie formula (1 kcal/mL), n (%)	858	54 (8.7)	20 (8.4)	0.89
Feeding interruption #, n (%)	858	85 (13.7)	33 (13.8)	1.0

Abbreviations: LE, lipid emulsion; * statistically significant at 5% level; # feeding interruption, if TPN was resumed and infant was kept NPO.

**Table 3 nutrients-14-03952-t003:** Univariate analysis of growth anthropometrics, neonatal morbidity, and mortality stratified by type of LE.

Variable	N	Intralipid (n = 620)	SMOFlipid (n = 238)	*p* Value
Growth anthropometrics				
Δ Weight z-score, median (IQR)	858	−1.2 (−2.26–−0.31)	−1.70 (−3.01–−0.57)	<0.001 *
Weight gain velocity (g/kg/day), median (IQR)	858	6.2 (3.6–8)	5.8 (2–7.7)	0.02 *
Weight at discharge # (g), median (IQR)	858	1820 (1402.5–1967.5)	1780 (1190–2000)	0.10
Weight at discharge # z-score, median (IQR)	858	−1.72 (−2.76–−1.07)	−1.94 (−2.86–−1.29)	0.02 *
HC at discharge # (g), median (IQR)	858	32 (31–34)	33 (32–35)	0.55
HC at discharge # z-score, median (IQR)	858	−1.2 (−1.82–−0.44)	−1.3 (−2.01–−65)	0.24
Δ HC z-score, median (IQR)	858	−1.02 (−1.69–−0.39)	−1.1 (−1.82–−0.53)	0.19
Neonatal morbidities and mortality				
Bronchopulmonary dysplasia, n (%)	582	209 (47.9)	44 (30.1)	<0.001 *
Late onset of sepsis (culture-proven), n (%)	858	227 (36.6)	122 (51)	<0.001 *
Retinopathy of prematurity stage ≥ 2, n (%)	575	43 (10.3)	11 (7.1)	0.27
Any intraventricular hemorrhage, n (%)	858	223 (36)	76 (31.8)	0.26
Severe intraventricular hemorrhage, n (%)	858	108 (17.4)	36 (15.1)	0.48
Necrotizing enterocolitis (stage ≥ 2), n (%): Medical management Surgical management	858	171 (27.6) 39 (6.3)	52 (21.8) 14 (5.9)	0.08 0.88
Spontaneous intestinal perforation, n (%)	858	5 (0.8)	5 (2.1)	0.15
Length of hospital stay, median (IQR)	858	41 (23–64)	38 (16–64)	0.06
Mortality, n (%)	858	134 (21.6)	63 (26.5)	0.15

Abbreviations: LE, lipid emulsion; * statistically significant at 5% level; # at discharge, at 36-week PMA or at discharge, whichever came first.

**Table 4 nutrients-14-03952-t004:** Multivariable regression for infant’s outcome stratified by type of LE.

	Unadjusted 95% CI	Adjusted 95% CI
Δ Weight z-score (MD)	−0.48 (−0.75, −0.20)	**−0.67 (−0.96, −0.39)**
Weight at discharge z-score # (MD)	2.07 (−9.08, 13.22)	−0.96 (−13.37, 11.46)
Bronchopulmonary dysplasia (RR)	0.63 (0.48, 0.82)	**0.61 (0.46, 0.80)**
Late-onset sepsis (RR)	1.40 (1.19, 1.65)	**1.44 (1.22, 1.69)**
Retinopathy of prematurity stage ≥ 2 (RR)	0.69 (0.37, 1.31)	0.43 (0.15–1.22)
Severe intraventricular hemorrhage (RR)	0.87 (0.61–1.22)	0.82 (0.55–1.22)
Necrotizing enterocolitis (RR)	0.76 (0.59–0.97)	0.78 (0.59–1.01)
Length of hospital stay (RR)	−6.59 (−13.0, 0.21)	−3.09 (−8.18, 2.00)
Mortality (RR)	1.23 (0.95–1.59)	1.21 (0.9–1.64)

Abbreviations: MD, mean difference, RR, relative risk, CI, confidence interval, LE, lipid emulsion; statistically significant at 5% level; # at discharge, at 36-week PMA or at discharge, whichever came first. Bold data is highlighted.

**Table 5 nutrients-14-03952-t005:** Effect of SMOFlipid on neonatal growth profile at 36 weeks or discharge.

Variable	Mean Confidence Interval		Mean Difference Confidence Interval	*p* Value	Interaction *p* Value
	Intralipid	SMOFlipid			Sex
Δ Weight z-score					0.001 *
Male	−1.13 (−1.33–−0.93)	−1.96 (−2.30–−1.64)	0.84 (0.46–1.22)	<0.001 *	
Female	−1.78 (−1.99–−1.56)	−1.85 (−2.19–−1.53)	0.08 (−0.31−0.47)	0.823	
Weight at discharge # (g)	1757.78 (1705.42–1810.13)	1668.57 (1586.53–1750.62)	89.21 (−9.18–187.59)	0.07	0.74
Weight at discharge z-score #	0.91 (−4.77–6.59)	2.98 (−7.31–13.27)	−2.07 (−13.8–−9.08)	0.72	0.77
Weight gain velocity (g/kg/day)	5.91 (5.07–6.75)	4.63 (3.94–5.31)	1.29 (−0.15–2.72)	0.08	0.12

* Statistically significant at 5% level; # at discharge, at 36-week PMA or at discharge, whichever came first.

## Data Availability

The data presented in this study are available on request from the corresponding author.

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
