# Peer review of "SMOFlipid Impact on Growth and Neonatal Morbidities in Very Preterm Infants"

_nutrients, 2022, doi:10.3390/nu14193952_

Round 1

Reviewer 1 Report

1.     Line 27:  Please define “intralipid” in abstract, probably in Line 17?  Should the first letter be capitalized as a trade name, i.e., Intralipid?  See :Line 31.

2.     Line 33:  It would be very helpful for the reader if you provided your opinion of what your data means, what does the reader do with your data, is Intralipid preferred in your opinion?

3.     Line 64:  Are you referring here to a specific formulation – “Intralipid” (trade name formulation) or to IV soy lipids here?  Please be specific here and throughout your manuscript.  

4.     Lines 95-98:  Please be specific as to exactly what data did you extract from patient charts.  Be specific as to what data, when, i.e. daily/weekly lipid profiles, daily/weekly body weights; who obtained body weights; were scales calibrated, etc.  Was it just 2 body weight comparisons birth and at 36 weeks?  How was GA determined? Did you obtain amounts for ALL non-IV exogenous nutrition/lipid sources?

You begin to provide some of this information in sections 2.5.1 through 2.6. If you feel this is better refer the reader to these sections for the listing of specific parameters collected.  However, I do see in these sections definitions of everything and most importantly the non-IV nutrition or additional IV fluid therapy that may also provide caloric support e.g., dextrose hydration fluids. 

5.     Line 101:  Does this mean the last weight could have been obtained at discharge where the infant is beyond 36 weeks PMA?  Same for Lines 105 and 135.

6.     Line 102:  PN was already defined in Line 39. 

7.     Line 113-114:  Please provide the % amino acid and the calcium concentration for your starter PN.

8.     Lines 121-122:  Is this 0.5 to 1gm/Kg per day?  Also, did every infant have their dose increased to 3.5 gm/Kg/day?  Please be specific. 

9.     Lines 144-146:  If I use this equation what do I determine?  What result do I obtain using this formula?   

10.  Table 1:  Are there data for “SGA”?  What defines “inotropes”?  

11.  Table 2:  Are the time periods shown for nutrition duration days?  Please state such.  Similarly, I assume the age for Enteral Nutrition is in days?  Should this be stated here?  As I note above in comment 4 above, would the amount of this enteral nutrition impact your Results?   What is feeding interruption?  What constitutes “High calorie formula”?

12.  Table 2:  I am confused as to what you mean by “Total PN Lipid/Protein intake” presented as g/Kg/day. How do I determine the total amount patients received?

13.  Tables 3, 4 and 5:  I remain confused as to what defines “at discharge”.  Please specifically define in Methods and include in Table footnotes.  Does your data analysis account for different total amounts/different days to discharge?

14.  Lines 292-298:  Should the non-accountability for the impact of non-IV nutrition calories/fat be included in Limitations?

15.  Lines 300-302:  Should you make a comment here regarding which lipid formulation your data suggests should be used based on weight gain, HC etc.?

Reviewer 2 Report

31: Define BPD

37: Need to define ‘very preterm infants.’

187-197: The sample size is extremely higher in the Intralipid cohort (72%) which could affect the final findings or p-value. The rational approach is to compare groups with an equal number of participants. This point should be addressed in the limitation section.

301: The finding related to postnatal growth failure is not clear enough in the results. How was it assessed?

300-302: Conclusion needs to be further improved. There is no mention of very preterm infants.

Clinical implications of the research need to be discussed in the discussion section.

Reviewer 3 Report

The article by Suzan S. Asfouret et al. “SMOFlipid impact on growth and neonatal morbidities in very preterm infants” is devoted to the characteristics of a lipid emulsion, containing the soya-bean oil, medium-chain triglycerides, olive oil, and fish oil lipid (SMOFlipid) as a parenteral nutrition for the preterm infants. The authors examined the effect of SMOFlipids which is used worldwide now, on growth velocity and neonatal morbidities in very preterm infants. Very preterm infants are born with insufficient energy. Parenteral nutrition (PN) is the main early source of energy and is the standard of care for preterm infants in the first days of life, so the detailed testing of its effect on the infant health is very important. In this article the impact of SMOFlipid on growth velocity, major neonatal outcomes, and preterm infants’ mortality is detailed tested and good presented. Nutritional and growth characteristics of study participants, analysis of growth anthropometrics, neonatal morbidity, and mortality are described and reviewed with a good sample size. The authors consider the existing contradictions in the current practice of using SMOFlipid in preterm infant. The article ia devoted to an actual medical problem, contains new reliable and signifacant data on the results of use of SMOFlipid as nutritional lipid emulsion for preterm infants and can be published in its present form.

Remarks. It is desirable to finalize the capture under figure 2: to explain all the curves and all the numbers in the picture.

Round 2

Reviewer 2 Report

Thank you for addressing the comments.